# Dynein Light Chain Protein Tctex1: A Novel Prognostic Marker and Molecular Mediator in Glioblastoma

**DOI:** 10.3390/cancers13112624

**Published:** 2021-05-27

**Authors:** Claudia Alexandra Dumitru, Eileen Brouwer, Tamina Stelzer, Salvatore Nocerino, Sebastian Rading, Ludwig Wilkens, Ibrahim Erol Sandalcioglu, Meliha Karsak

**Affiliations:** 1Department of Neurosurgery, Otto-von-Guericke University, 39120 Magdeburg, Germany; erol.sandalcioglu@med.ovgu.de; 2Center for Molecular Neurobiology (ZMNH), University Medical Center Hamburg-Eppendorf (UKE), 20246 Hamburg, Germany; E.M.Brouwer-12@umcutrecht.nl (E.B.); 18IMC10213@fh-krems.ac.at (T.S.); s257231@studenti.units.it (S.N.); sebastian.rading@zmnh.uni-hamburg.de (S.R.); 3Department of Pathology, Nordstadt Hospital Hannover, 30167 Hannover, Germany; ludwig.wilkens@krh.eu

**Keywords:** glioblastoma, Tctex1/DYNLT1, prognostic biomarkers, tumor proliferation, tumor invasion

## Abstract

**Simple Summary:**

Glioblastoma (GBM) remains one of the deadliest solid cancers, with only a dismal proportion of GBM patients achieving 5-year survival. Thus, it is critical to identify molecular mechanisms that could be targeted by novel therapeutic approaches in this tumor type. Our study identified Tctex1/DYNLT1 as an independent prognostic marker for the overall survival of GBM patients. Importantly, Tctex1 promoted the aggressiveness of GBM cells by enhancing tumor proliferation and invasion. These effects of Tctex1 appeared to be modulated via phosphorylation of retinoblastoma protein (RB) and the release of matrix metalloprotease 2 (MMP2), respectively. As Tctex1 can potentially be inhibited in vivo, our study provides a rationale for novel, individualized therapeutic strategies in GBM patients.

**Abstract:**

The purpose of this study was to determine the role of Tctex1 (DYNLT1, dynein light chain-1) in the pathophysiology of glioblastoma (GBM). To this end, we performed immunohistochemical analyses on tissues from GBM patients (*n* = 202). Tctex1 was additionally overexpressed in two different GBM cell lines, which were then evaluated in regard to their proliferative and invasive properties. We found that Tctex1 levels were significantly higher in GBM compared to healthy adjacent brain tissues. Furthermore, high Tctex1 expression was significantly associated with the short overall- (*p* = 0.002, log-rank) and progression-free (*p* = 0.028, log-rank) survival of GBM patients and was an independent predictor of poor overall survival in multivariate Cox-regression models. In vitro, Tctex1 promoted the metabolic activity, anchorage-independent growth and proliferation of GBM cells. This phenomenon was previously shown to occur via the phosphorylation of retinoblastoma protein (phospho-RB). Here, we found a direct and significant correlation between the levels of Tctex1 and phospho-RB (Ser807/801) in tissues from GBM patients (*p* = 0.007, Rho = 0.284, Spearman’s rank). Finally, Tctex1 enhanced the invasiveness of GBM cells and the release of pro-invasive matrix metalloprotease 2 (MMP2). These findings indicate that Tctex1 promotes GBM progression and therefore might be a useful therapeutic target in this type of cancer.

## 1. Introduction

Glioblastoma (GBM) is the most common and aggressive malignant brain tumor in adults, with an incidence of 0.59–3.69 cases per 100,000 person life-years [1]. The vast majority of GBM develop de novo (primary GBM) and occur more commonly in male patients [2]. The 5-year relative survival of GBM patients is less than 10%, even under multimodal therapeutic approaches consisting of the maximum safe surgical resection followed by radiation therapy with concomitant Temozolomide chemotherapy [1,2,3]. Although a large body of research has addressed signaling pathways, biomarkers and potential therapeutic targets in GBM, this disease remains incurable at present. Thus, there is an urgent need to identify novel cellular and molecular mechanisms that are involved in the pathophysiology of GBM, thereby controlling its progression.

Homodimeric Tctex1 (DYNLT1, dynein light chain-1) is part of the dynein motor complex (cytoplasmic dynein-1) [4,5] and interacts with dynein intermediate chains (DIC), using its two symmetrical, hydrophobic grooves [6]. Cytoplasmic dynein-1 is the most versatile, microtubule-based motor in eukaryotic cells, as it is able to transport many different cargoes, including organelles, vesicles, RNAs, protein complexes and even viruses (reviewed in [7]). This versatility is underscored by the fact that animal cells contain a single dynein, but around 40 kinesins that perform related functions [7]. Apart from intracellular transport, cytoplasmic dynein is also an important modulator of the cell cycle because it can control centrosome separation, nuclear envelope breakdown, spindle assembly, checkpoint inactivation, chromosome segregation and spindle positioning (reviewed in [8]).

Previous studies showed that Tctex1 itself could directly regulate the length of the G1-phase and induce S-phase entry, thereby promoting cell cycle progression and proliferation of neural progenitor cells [9]. This phenomenon seemed to be, however, dynein-independent and could be attributed to the pool of Tctex1 that existed outside the dynein complex [10,11,12,13]. It has also been shown that dynein-independent Tctex1 selectively activated heterotrimeric G proteins in the absence of a typical G protein-coupled receptor (GPCR) by interacting with the β- and γ-subunits [14]. Gβγ isoforms competed with DIC for Tctex1 binding, thereby regulating its assembly into the dynein motor complex. While binding to DIC promoted the incorporation of Tctex1 into the dynein motor complex, dynein-free Tctex1 bound to Gβγ caused neurite outgrowth [15]. A similar competition mechanism has been shown to mediate RhoGEF GEF-H1 activation downstream of GPCR stimulation. Specifically, G-protein subunits promoted the disassembly of the GEF-H1-Tctex1-DIC complex and the displacement of GEF-H1 from the microtubule array. The collaboration of Gα and Gβγ by competing with the GEF-H1-Tctex1 and Tctex1-DIC interfaces, respectively, resulted in GEF-H1 activation [16].

The role of Tctex1 in GBM is currently unknown. This study aims to determine (1) the association between Tctex1 expression levels in GBM tumors and the patients’ clinical outcome and (2) whether Tctex1 modulates major biological functions of GBM cells such as tumor proliferation and invasion.

## 2. Materials and Methods

### 2.1. Study Subjects

In this study, we retrospectively analyzed tissues from 202 adult patients with histopathologically confirmed, newly diagnosed GBM. The tumors were clinically classified as primary GBM, as no lower grade glioma had been documented in the patients’ medical history. All patients were treated at the Department of Neurosurgery, Nordstadt Hospital Hannover between 2004 and 2014 and had a median age of 66 years. The studies were carried out in accordance with the Declaration of Helsinki of 1975, revised in 2013. The Ethics Committee of the Medical School Hannover approved these studies and provided a waiver for the need for informed consent (Study Nr. 6864, 2015). The clinical characteristics of the patients including sex, post-operative Karnofsky Performance Scale (KPS), therapy, extent of surgical resection, MGMT methylation and IDH mutation status are summarized in Table 1. From a subgroup of patients (*n* = 42), we were also able to retrieve adjacent, tumor-free brain tissues, which were subsequently used as “healthy” controls in our immunohistochemical studies.

### 2.2. Tissue Microarrays (TMA): Construction, Immunohistochemistry and Scoring

TMA blocks were built using the Arraymold kit E (Riverton, UT, USA), as previously described [17]. Briefly, cores derived from vital and solid tumor tissue areas were extracted from formalin-fixed/paraffin-embedded (FFPE) GBM tissues, using a 3 mm biopsy punch. The cores were transferred into recipient blocks and cut into 2 µm sections. Prior to staining, the sections were deparaffinized, and the antigens were retrieved by heat-induced antigen retrieval (HIER) in citrate buffer pH 6.0 (Thermo Scientific, Freemont, CA, USA). The sections were stained with 406 ng/mL polyclonal rabbit anti-Tctex1 antibodies (Proteintech Europe, Manchester, UK), 1 µg/mL monoclonal rabbit anti-phospho-RB (Ser807/811) antibodies (Cell Signaling Technology, Frankfurt am Main, Germany) or 19 µg/mL monoclonal mouse anti-RB antibodies (Cell Signaling Technology) overnight at 4 °C. Secondary and colorimetric reactions were performed using the UltraVisionTM LP Detection System, according to the manufacturer’s instructions (Thermo Scientific). Nuclei were counterstained with hematoxylin (Carl Roth, Karlsruhe, Germany), and the sections were covered with Mountex^®^ embedding medium (Medite, Burgdorf, Germany). All stained TMAs were digitalized with an Aperio AT2 high-resolution, whole-slide scanner, and the digital images were viewed with Aperio ImageScope software (Leica Biosystems, Nussloch, Germany). Blinded histological analysis was performed independently by authors C.A.D., M.K. and L.W. (senior histopathologist).

Tctex1 exhibited, mainly, a cytoplasmic subcellular localization. The expression intensity of Tctex1 was categorized as ‘’weak“, ‘’medium“ or ‘’strong“ and was assigned 1, 2 or 3 points, respectively (Figure 1A). As a number of samples exhibited heterogenous staining, Tctex1 expression was subsequently graded using the H-Score according to the formula: (1 × X) + (2 × Y) + (3 × Z), where X + Y + Z = 100% of the total tumor area. Phospho-RB, which exhibited exclusively a nuclear localization, was assessed as percentage of positive cells using a 5-tier scoring system (Figure 1B). At least two different fields per TMA spot were analyzed at 200-fold magnification, and the values were subsequently averaged. Total RB exhibited mostly nuclear and some cytoplasmic subcellular localization. The staining pattern was mostly homogenous, and the samples were subsequently categorized as ‘’weak/negative” or ‘’positive” (Figure 1C).

### 2.3. Cell Lines and Stable Transfections

Both GBM cell lines used in this study (U87-MG and U373-MG) are commercially available. The U87-MG cells were purchased from Sigma-Aldrich, St. Louis, MO, USA (Cat. 89081402); U373-MG were a kind gift from Prof. K. Lamszus (University Medical Center Hamburg-Eppendorf, Germany). Both cell lines were cultured in Dulbecco’s Modified Eagle Medium (Gibco^®^ DMEM; ThermoFisher Scientific, Rockford, IL, USA) supplemented with 10% fetal calf serum (FCS; Pan Biotech, Aidenbach, Germany), 1% Penicillin-Streptomycin and 1% non-essential amino acids (both from Sigma-Aldrich). The cells were transfected with pcDNA3.1(+) Tctex1 or pcDNA3.1(+) plasmids using Lipofectamine^®^2000 (Life Technologies, ThermoFisher Scientific) in Opti-MEM^®^ medium (Gibco, ThermoFisher Scientific), according to the manufacturer’s protocol. Transfected cells were selected with 400 µg/mL G418 (InvivoGen, Toulouse, France) and then maintained in cell culture medium containing 200 µg/mL G418.

### 2.4. SDS-PAGE and Western Blot

GBM cells were lysed in an ice-cold buffer containing 0.2% DDM (N-Dodecyl-beta-Maltoside) and protease/phosphatase inhibitors (both from Sigma-Aldrich). The Pierce BCA Protein Assay kit (ThermoFisher Scientific) was used to determine the protein concentration in the DDM extracts. Cell debris was removed by centrifugation, and the lysates were incubated with an SDS-Loading buffer containing DTT (Sigma-Aldrich) at a final concentration of 16.6 mM. Samples were separated by SDS-PAGE, followed by transfer to nitrocellulose membranes (GE Healthcare Amersham, Chalfont Saint Giles, UK). The membranes were incubated with 8 µg/mL mouse anti-Tctex1 antibodies (Sigma-Aldrich) or 1 µg/mL Calnexin antibodies (Enzo Life Sciences, Lörrach, Germany) overnight at 4 °C, followed by IRDye^®^-coupled secondary antibodies (LI-COR Biosciences, Bad Homburg, Germany) for 1h at room temperature. Signal detection was performed on a LI-COR Oddyssey^®^ CLx imaging system.

### 2.5. MTT Assay

GBM cells were seeded at a concentration of 8000 cells/well in 96-well plates. At the indicated time points, fresh medium containing 10% MTT (3-(4,5-dimethylthiazol-2-yl)-2,5-diphenyltetrazolium bromide) (ThermoFisher Scientific) was added, and the samples were incubated for another 3 h (U373 cells) or 4 h (U87 cells) at 37 °C to allow formation of formazan crystals. After solubilization, colorimetric detection was performed at OD_550_-OD_690_ on a TECAN plate reader (Tecan, Männedorf, Switzerland). Each biological replicate (experiment) included 36 (for U373) and 46 (for U87) technical replicates per plate. Subsequently, the mean value of the replicates was calculated for pcDNA and pcDNA Tctex1 samples in each cell line. The mean blank value consisted of four technical replicates per plate and was subtracted from the measured values per well. The experiments were performed independently three times (for U373) and four times (for U87).

### 2.6. Soft-Agar Clonogenic Assay

24-well plates were coated with 1% high-gelling agarose (PanReac Applichem, Darmstadt, Germany). GBM cells were mixed with low-gelling agarose (Carl Roth, Karlsruhe, Germany) at a final concentration of 0.3% and were added on top of the first layer. The low-gelling agarose was allowed to solidify for 1 h at 4 °C, and culture medium was added in each well. The samples were incubated at 37 °C for 15 days, with medium change every 3–4 days. The colonies were stained with a solution containing 0.005% Crystal Violet (Sigma-Aldrich) and counted on a SZX16 microscope (Olympus, Hamburg, Germany). Each experiment contained 12 technical replicates per cell line and transfection condition, which were subsequently averaged. The experiments were performed independently three times (for U373) and four times (for U87).

### 2.7. Immunofluorescence Assay

GBM cells were seeded on poly-L-Lysine treated coverslips (Sigma-Aldrich). The samples were fixed with a solution containing 3.7% PFA and 1% sucrose and then permeabilized with 0.2% Triton-X-100 (Sigma-Aldrich). The cells were stained with 4 µg/mL Ki67 antibodies (ThermoFisher Scientific) overnight at 4 °C followed by AlexaFluor488-coupled secondary antibodies (Life Technologies) for 1 h at room temperature. The cells were counterstained with TexasRed^®^-X-Phalloidin (ThermoFisher Scientific) and alpha-Tubulin (Sigma-Aldrich), followed by AlexaFluor647-coupled secondary antibodies (Invitrogen). The samples were embedded in Prolong^TM^ Diamond antifade medium (Invitrogen) and analyzed on a Zeiss Imager M2 LSM900 fluorescence microscope (Carl Zeiss, Oberkochen, Germany). The mean signal value was measured with ImageJ Version JDK8 (https://imagej.nih.gov/ij/, accessed on 18 March 2021) software. Each experiment contained three technical replicates per cell line and transfection condition. Up to three images were acquired for each technical replicate, resulting in a maximum of nine images per sample. The experiments were performed independently, five times for each cell line.

### 2.8. Gelatin Zymography

The release of matrix metalloproteases (MMPs) by GBM cells was analyzed by gelatin zymography, as described previously [18]. Briefly, 10^5^ cells/mL were incubated at 37 °C in DMEM medium, supplemented as above. As serum-supplemented cell culture medium also contained MMPs, medium without cells was used as control. The supernatants were collected at 24 h (for the U87 cells) or 48 h (for the U373 cells) and mixed with Zymogram sample buffer at a final concentration of 80 mM Tris pH 6.8, 1% SDS, 4% glycerol and 0.006% bromophenol blue. Proteins were separated by SDS-PAGE containing 0.2% gelatin 180 Bloom and then renatured in 2.5% Triton-X-100 for 1 h at room temperature. The enzymatic reaction was performed overnight at 37 °C in a buffer containing 50 mM Tris pH 7.5, 200 mM NaCl, 5 mM CaCl_2_ and 1% Triton-X-100. The gels were stained with a solution containing 0.5% Coomassie blue, 30% methanol and 10% acetic acid for 1 h at room temperature. Finally, the gels were de-stained with 30% methanol and 10% acetic acid until the digested bands became visible. All chemicals were from Carl Roth (Karlsruhe, Germany). The gelatinolytic bands were quantified with ImageJ 1.48v software (https://imagej.nih.gov/ij/, accessed on 18 March 2021). The experiments were performed, independently, three times for each cell line.

### 2.9. Invasion Assay

The invasion of GBM cells was assessed with the ORIS^TM^ cell invasion system (Platypus Technologies LLC, Madison, WI, USA), according to the manufacturer’s instructions. The U87 cells were allowed to invade for 24 h in a matrix containing 2 mg/mL collagen I; the invasion of U373 cells was assessed at 72 h in a matrix containing 1 mg/mL collagen I. The degree of “gap-closure” was quantified with ImageJ 1.48v software. The experiments were performed, independently, three times for each cell line.

### 2.10. Statistical Analysis

Clinical data was analyzed with SPSS statistical software version 26 (IBM Corporation). Survival curves (5-year or 1-year cut-off) were plotted according to the Kaplan–Meier method. Significance was initially tested by univariate analysis, using the log-rank test. Multivariate analysis was subsequently used to determine the prognostic value of selected variables using Cox’s proportional hazard linear regression models, adjusted for age, KPS, therapy, extent of surgical resection, MGMT methylation and IDH mutation status. The difference in marker expression between GBM and healthy tissues was analyzed using box–whisker plots. Statistical testing in matched tumor–healthy tissue samples was subsequently performed using the Wilcoxon test. Correlation analysis was performed with Spearman’s rank test (Spearman’s rho). The data obtained with GBM cell lines was analyzed with the two-tailed Student’s *t*-test. In all studies, the level of significance was set at *p* ≤ 0.05.

## 3. Results

### 3.1. Tctex1 in Glioblastoma (GBM) Patients

In the first set of studies, we determined whether Tctex1 was differentially expressed in GBM tissues compared to their tumor-free, adjacent brain tissue counterparts. The expression levels of Tctex1 were displayed as box–whisker plots, and the significance was tested with the Wilcoxon test. The results showed that Tctex1 was significantly increased in GBM compared to tumor-free brain tissue (*p* < 0.001; Wilcoxon) (Figure 2A). Representative examples of Tctex1 expression in GBM versus tumor-free tissues are shown in Figure 2B—left and right panels, respectively. The middle panel depicts the border area, where the differential expression of Tctex1 in a tumor versus adjacent tumor-free brain is clearly noticeable (Figure 2B). Micrographs with higher magnification and resolution are provided in Appendix A.

We next investigated whether Tctex1 might associate with overall survival or progression-free survival in GBM patients. To this end, we dichotomized the expression levels of Tctex1 into “low” and “high”, using as a cut-off the median values of the tumor-free, adjacent brain tissues. The rationale behind this method of dichotomization was that healthy brain tissues expressed baseline, physiological levels of Tctex1, while an overexpression of Tctex1, as seen in many GBM tumors, was likely linked to a pathological process. The survival curves were plotted according to the Kaplan–Meier method, and statistical significance was assessed with the log-rank test. We found that GBM patients with high levels of Tctex1 had a significantly shorter overall survival (*p* = 0.002; log-rank) and progression-free survival (*p* = 0.028; log-rank) compared to patients with low levels of Tctex1 (Figure 2C,D).

The survival data were further analyzed using a Cox proportional-hazard model, adjusted for factors that were known to influence the outcome of GBM patients such as age, Karnofsky Performance Scale (KPS), therapy, extent of surgical resection, MGMT methylation and IDH mutation status [19,20,21,22]. The proportional hazard assumption testing for Tctex1 showed a *p*-value of 0.515, indicating that the assumption of this model was satisfied. We found that Tctex1 was a significant predictor of poor overall survival (HR = 1.67, 95% CI = 1.08–2.58, *p* = 0.021) but not of progression-free survival (HR = 1.54, 95% CI = 0.83–2.86, *p* = 0.167) in GBM patients (Figure 2E). These data indicate that Tctex1 could serve as an independent prognostic biomarker, at least for the overall survival of patients with this type of cancer.

### 3.2. Tctex1 and GBM Proliferation

To assess the role of Tctex1 in GBM biology and functions, we first established an in vitro model of Tctex1 overexpression. To this end, two GBM cell lines (U373 and U87) were stably transfected to overexpress Tctex1. A representative example of Tctex1 protein levels in GBM cells transfected with the overexpression plasmid (pcDNA Tctex1) versus control plasmid (pcDNA) are shown in Figure 3A,B. The corresponding original western blots are shown in Appendix A, respectively.

Next, we determined the metabolic activity of transfected GBM cells, using the MTT assay. The results showed that Tctex1 overexpression significantly increased metabolic activity in both U373 and U87 cells, starting at 3 days in culture (Figure 4A,B). We additionally determined the anchorage-independent growth of transfected GBM cells by allowing the cells to form colonies in low-gelling agarose for 15 days (Figure 4C). We found that both U373 and U87 pcDNA Tctex1 cells formed significantly more colonies than their pcDNA counterparts (Figure 4D). Finally, we assessed the staining intensity of the specific proliferation marker, Ki67, by immunofluorescence (Figure 4E). The results showed that Tctex1-overexpressing cells had significantly higher Ki67 positivity compared to control in both U373 and U87 cell lines (Figure 4F). Taken together, these data indicate that Tctex1 promotes the proliferation of GBM cells.

Previous studies proposed that the effect of Tctex1 on cell cycling and proliferation was mediated via phosphorylation of the retinoblastoma protein (RB) [9]. To test this hypothesis, we stained TMAs from GBM patients with specific antibodies against the phosphorylated form of RB. The samples were, additionally, stained against the total form of RB. The expression levels of phospho-RB in matched GBM–healthy brain tissue pairs were displayed as box–whisker plots, and the significance was tested with the Wilcoxon test. Similar to Tctex1, phospho-RB was significantly increased in GBM compared to the tumor-free brain tissues (*p* = 0.001; Wilcoxon) (Figure 5A). Representative examples of phospho-RB expression in GBM versus tumor-free tissues are shown in Figure 5B—left and right panels, respectively—while the middle panel depicts the border area (Figure 5B). To determine the relationship between phospho-RB and Tctex1 in GBM patients (*n* = 108), we performed correlation analysis using Spearman’s rank test. The results showed that phospho-RB directly and significantly correlated with Tctex1 in these tissues (*p* = 0.007, Rho = 0.284, Spearman’s rank). In contrast, the total levels of RB did not correlate with Tctex1 in these patients (*p* = 0.836, Rho = 0.020, Spearman’s rank). Representative examples of synchronous low and high levels of Tctex1/phospho-RB are shown in Figure 5C,D, respectively (upper and middle panels). The lower panels depict the expression levels of total RB in these samples (Figure 5C,D).

### 3.3. Tctex1 and GBM Invasion

In the last set of studies, we tested whether Tctex1 might regulate the invasiveness of GBM cells. The invasion of the tumor cells was assessed by the degree of “gap” closure (red line) in a 3D collagen matrix, using the Oris^TM^ system (Figure 6A,B). The results showed that Tctex1 overexpression significantly increased the invasiveness of both U373 and U87 GBM cells (Figure 6C).

As matrix metalloproteases (MMPs) were known to promote the invasion of GBM cells [23,24,25,26], we also determined the activity of released MMPs using gelatin zymography. To this end, pcDNA and pcDNA Tctex1 GBM cells were incubated in culture medium, and the supernatants were collected at 48 h (U373) or 24 h (U87). Culture medium without cells was used as control. We found that both U373 and U87 cells released relatively high levels of MMP2, but only negligible levels of MMP9 (Figure 6D). Importantly, Tctex1-overexpressing GBM cells had significantly higher levels of MMP2 compared to their control-transfected counterparts (Figure 6E,F). Together, these findings indicate that Tctex1 enhances GBM invasiveness, possibly via MMP2.

## 4. Discussion

Increased effort has been made to identify cellular/molecular factors that modulate the pathophysiology of GBM and could provide information regarding diagnosis, prognosis and potential therapeutic approaches for this type of cancer. Our study is the first to identify Tctex1/DYNLT1 as a tumor-promoting factor in GBM that may furthermore serve as a novel independent prognostic marker for the overall survival of GBM patients.

Our results showed that Tctex1 was overexpressed in GBM compared to healthy brain tissues. Although the molecular mechanisms mediating Tctex1 overexpression in GBM are currently unknown, previous studies identified several upstream modulators of Tctex1 in other systems. For instance Fang et al. showed that MAP4 overexpression in HeLa cells resulted in an elevated expression of Tctex1, while Huo et al. found that knock-down of vimentin in ovarian cancer cells induced changes in the mRNA expression of Tctex1 [27,28]. Additionally, Wie et al. demonstrated that high levels of exosomal miR-15b-3p mRNA downregulated the expression of Tctex1 in gastric cancer cell lines [29]. Whether these mechanisms also modulate Tctex1 expression in GBM remains, however, to be determined in future studies. We further found that GBM patients with high levels of Tctex1 (Tctex1^high^) had a significantly shorter overall and progression-free survival compared to Tctex1^low^ patients. These findings prompted us to investigate whether Tctex1 might modulate the biology and functions of GBM cells.

The role of Tctex1 in cancer biology is poorly characterized at present. In fact, a very recent PubMed search using the keywords “Tctex1” or “DYNLT1” and “cancer” resulted in only 12 hits. In the present study, we found that Tctex1 increased the metabolic activity of GBM cells. Furthermore, Tctex1 promoted the anchorage-independent growth and the proliferation (as indicated by Ki67 positivity) of the tumor cells. These findings are supported by previous studies showing that Tctex1 regulated the length of the G1-phase, induced S-phase entry and, consequently, promoted cell cycle progression and cell proliferation [9,11]. This phenomenon was proposed to occur via phosphorylation of retinoblastoma protein (phospho-RB) because Tctex1 knock-down reduced the levels of phospho-RB and prevented S-phase entry in retinal pigmented epithelial cells [9]. Interestingly, our study found a direct and significant correlation between the levels of phospho-RB and those of Tctex1 in GBM tissues. We found, however, no significant correlation between Tctex1 and the total levels of RB, which indicated that the potential effect of Tctex1 on RB occurred via post-translational modifications. Thus, our data are in agreement with the previously described mechanism of Tctex1-induced RB phosphorylation [9] and suggest that the Tctex1-phospho-RB axis could modulate the cell cycle and proliferation in GBM cells as well.

We further found that Tctex1 induced the release of extracellular matrix-degrading MMP2 and enhanced the invasion of GBM cells. These findings are of particular importance for the pathophysiology of GBM because the high invasiveness of GBM cells is the main reason why this type of tumor cannot be successfully removed by surgery. The molecular mechanisms mediating this novel function of Tctex1 are currently unknown and remain to be elucidated in future studies. One possible explanation is provided, however, by the role of cytoplasmic dynein-1 as a critical microtubule-based motor, which allows it to transport a very diverse array of cargoes [7]. This hypothesis is supported by previous studies linking microtubule-dependent trafficking with the cellular release of MMPs. Specifically, Schnaeker at al. found that melanoma cells contained cytoplasmic vesicles positive for MMP2 and MMP9 that were released via microtubules to induce tumor invasion [30]. Furthermore, Sbai and colleagues showed that the intracellular transport of MMP9 in neuroblastoma cells was partly dependent on the dynein/dynactin molecular motor [31].

Taken together, these findings indicate that Tctex1 enhances the aggressiveness of GBM cells and promotes tumor progression. While a direct pharmacological inhibition of Tctex1 for therapeutic purposes is currently not available, the tumor levels of Tctex1 could be reduced in an indirect manner. Previous studies showed that the interaction of Tctex1 with DIC and several Gβ proteins is carried out via a putative binding domain K/R-K/R-X-X-X-K/R [11,15]. This binding motif is also found in the carboxyl terminal portion of cannabinoid receptor CB2. Studies from our group found that the cannabinoid receptor CB2 competed with Tctex1 in binding to Gβγ subunits, thereby disrupting the Tctex1-Gβγ interaction. As a result, Tctex1 was released from the complex and underwent degradation by proteasomes and, partly, by lysosomes [32]. Importantly, a number of studies—including phase I and II clinical trials—showed that activation of cannabinoid receptors by tetrahydrocannabinol (THC) or cannabidiol (CBD) had potent antineoplastic effects on GBM in vivo ([33] and reviewed in [34]). Thus, Tctex1 may be a potential target in GBM, and patients with high tumor levels of Tctex1 might benefit from individualized therapeutic approaches involving the cannabinoids–cannabinoid receptors axis.

## 5. Conclusions

In summary, this study identifies Tctex1 as an independent prognostic biomarker in GBM. Furthermore, we demonstrate that Tctex1 mediates important biological functions of GBM cells such as proliferation and invasion, potentially via phosphorylation of RB and MMP2 release, respectively. These findings contribute to a better understanding of GBM pathophysiology and might ultimately foster the development of novel therapeutic strategies against this type of cancer.

## Figures and Tables

**Figure 1 cancers-13-02624-f001:**
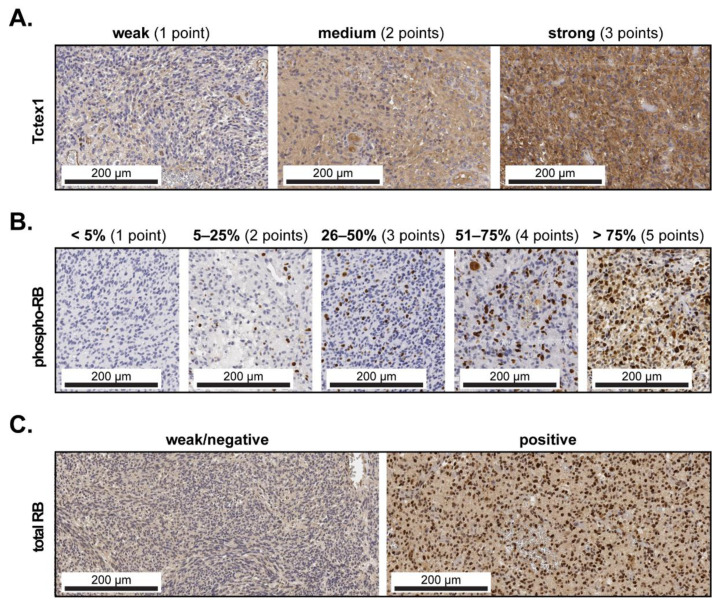
Marker expression and scoring in GBM tissues. (**A**) Representative micrographs showing weak (1 point), medium (2 points) and strong (3 points) expression of Tctex1 in GBM tissues. The H-score was subsequently calculated according to the formula: (1 × X) + (2 × Y) + (3 × Z), where X + Y + Z = 100% of the total tumor area. Representative micrographs showing (**B**) the 5-tier score for phospho-RB according to the percentage of positive cells and (**C**) weak/negative or positive staining for total RB. The scale bars are indicated in the lower-left corner of each panel.

**Figure 2 cancers-13-02624-f002:**
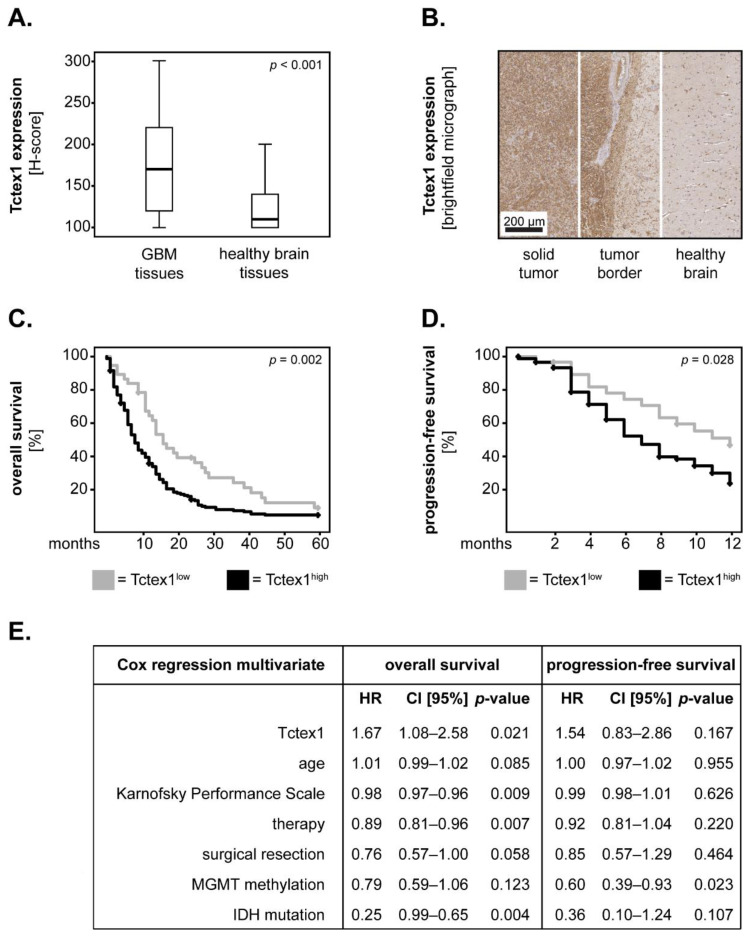
Tctex1 in GBM patients. (**A**) Tctex1 expression in matched pairs of GBM tissues and adjacent, tumor-free brain tissues (*n* = 42). The medians are shown as black lines and the percentiles (25th and 75th) as vertical boxes with error bars. Statistical analysis was performed with the Wilcoxon test. (**B**) Representative micrographs showing the expression of Tctex1 in the solid tumor area (left panel), tumor border area (middle panel) and adjacent, tumor-free tissue area (right panel), respectively. Kaplan–Meier curves for the (**C**) 5-year overall survival and (**D**) 1-year progression-free survival of GBM patients with high versus low levels of Tctex1. The log-rank test was used for statistical analysis, and the *p*-values are indicated in the upper-right corner of each plot. (**E**) Multivariate Cox regression analysis model for the overall and progression-free survival of patients with high versus low levels of Tctex1. HR: hazard ratio; CI [95%]: 95% confidence interval.

**Figure 3 cancers-13-02624-f003:**
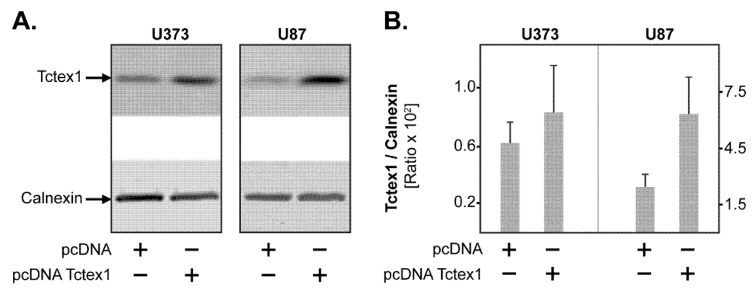
Tctex1 overexpression in GBM cell lines. U373 and U87 GBM cells were transfected with a Tctex1 overexpression plasmid (pcDNA Tctex1) or the control plasmid (pcDNA). After antibiotic selection, the cells were lysed and analyzed by Western blot, using Tctex1-specific antibodies. (**A**) Representative Western blot showing Tctex1 expression in pcDNA versus pcDNA Tctex1 cells. Calnexin was used as the loading control. (**B**) Tctex1/Calnexin ratio in pcDNA versus pcDNA Tctex1 GBM cells. Shown are the means + S.D. of 3 independent cell lysates.

**Figure 4 cancers-13-02624-f004:**
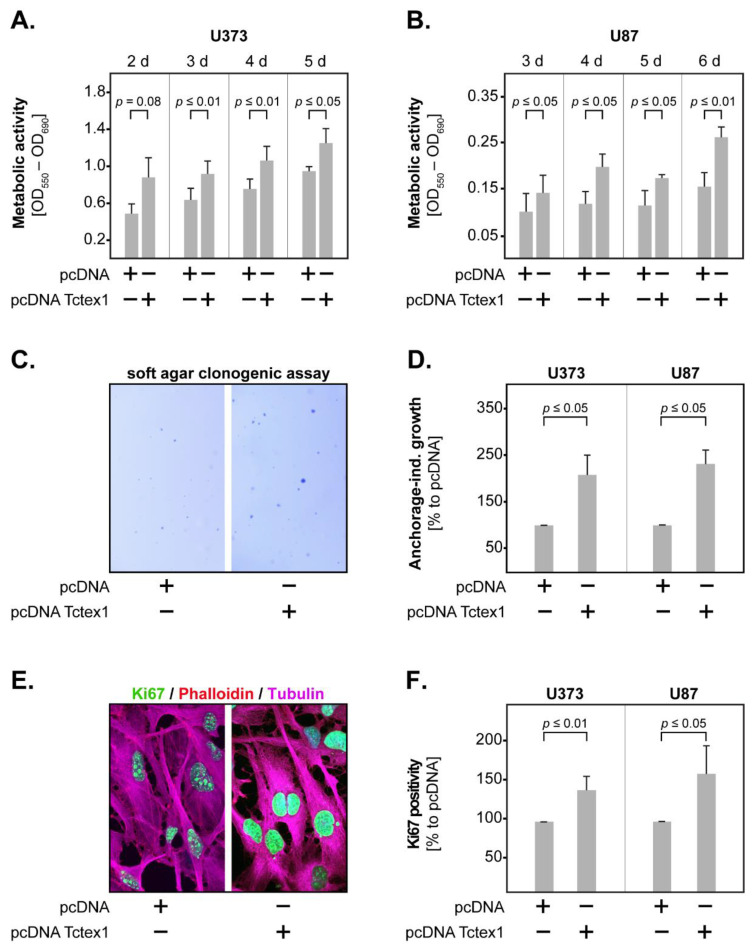
Tctex1 and GBM proliferation. Tctex1 overexpression significantly increased the metabolic activity of (**A**) U373 and (**B**) U87 GBM cells, as indicated by the MTT assay. (**C**) Representative micrographs of colonies generated by pcDNA and pcDNA Tctex1 U373 cells after 15 days of culture in low-gelling agarose. (**D**) Tctex1 overexpression significantly increased the anchorage-independent growth of U373 and U87 GBM cells. (**E**) Representative micrographs showing Ki67 staining (green) in pcDNA and pcDNA Tctex1 U373 cells. The cells were co-stained with phalloidin (red) and tubulin (magenta). (**F**) Tctex1 overexpression significantly increased Ki67 positivity in U373 and U87 GBM cells. At least 3 independent experiments were performed for each assay. In all studies, statistical analysis was performed with the paired *t*-test.

**Figure 5 cancers-13-02624-f005:**
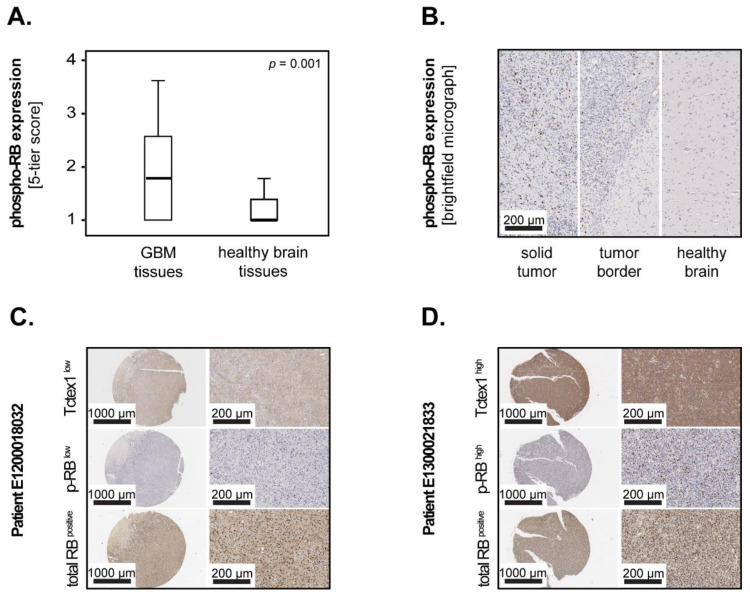
phospho-RB in GBM patients. (**A**) Expression of phospho-RB in matched pairs of GBM tissues and adjacent tumor-free brain tissues (*n* = 14). The medians are shown as black lines, and the percentiles (25th and 75th) as vertical boxes with error bars. Statistical analysis was performed with the Wilcoxon test. (**B**) Representative micrographs showing the expression of phospho-RB in the solid tumor area (left panel), tumor border area (middle panel) and adjacent tumor-free tissue area (right panel), respectively. Representative micrographs showing (**C**) synchronous low levels and (**D**) synchronous high levels of Tctex1 and phospho-RB in GBM tissues. The corresponding levels of total RB are shown in the lower panels.

**Figure 6 cancers-13-02624-f006:**
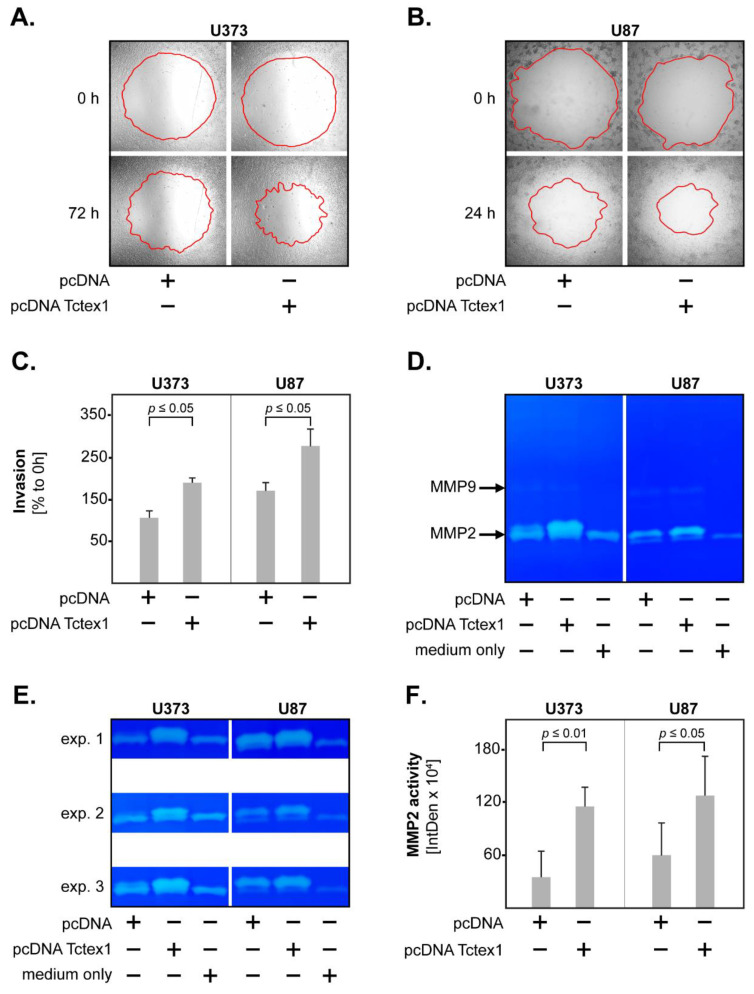
Tctex1 and GBM invasion. Representative micrographs of invasion assays in pcDNA versus pcDNA Tctex1 (**A**) U373 cells and (**B**) U87 cells. The upper panels show the pre-invasion status at 0 h; the lower panels show the post-invasion status at 72 h (for U373) or 24 h (for U87). The red lines mark the closure of the “gap“, indicating the degree of tumor invasion. (**C**) Tctex1 overexpression significantly increased the invasiveness of U373 and U87 GBM cells. The data are presented as percentage of pre-invasion status. Shown are the means + S.D. of 3 independent experiments. Statistical analysis was performed with the paired *t*-test. (**D**) Representative result of a gelatin zymography assay showing that both GBM cell lines released MMP2, but only negligible levels of MMP9. (**E**) Gelatinolytic activity of MMP2 in pcDNA versus pcDNA Tctex1 GBM cells. The results of 3 independent experiments are shown. (**F**) Tctex1 overexpression significantly increased the release of MMP2 in U373 and U87 GBM cells. The data are presented as IntDen (integrated density) values, from which the medium-only sample has been subtracted. Shown are the means + S.D. of 3 independent experiments. Statistical analysis was performed with the paired *t*-test.

**Table 1 cancers-13-02624-t001:** Clinical characteristics of GBM patients. RTX: radiotherapy; CTX: chemotherapy; RCTX: radio-chemotherapy; n.d.: not determinable.

All Patients	Number	Percentage
	202	100
**Sex**		
female	82	40.6
male	120	59.4
**Karnofsky Performance Scale (KPS)**		
10	1	0.5
20	1	0.5
30	1	0.5
40	9	4.5
50	29	14.4
60	44	21.8
70	40	19.8
80	39	19.3
90	30	14.9
100	1	0.5
n.d.	7	3.5
**Therapy**		
surgery	20	9.9
surgery + RTX	27	13.4
surgery + CTX	5	2.5
surgery + RCTX	140	69.3
n.d.	10	5.0
**Surgical resection**		
total	79	39.1
subtotal	110	54.5
n.d.	13	6.4
**MGMT methylation**		
unmethylated	92	45.5
methylated	96	47.5
n.d.	14	6.9
**IDH mutation status**		
wild-type	182	90.1
mutated	9	4.5
n.d.	11	5.4

## Data Availability

Data from the in vitro studies is contained within the article and Appendix A. The datasets involving GBM patients are available from the corresponding author on request.

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
