# Peer review of "Dynein Light Chain Protein Tctex1: A Novel Prognostic Marker and Molecular Mediator in Glioblastoma"

_cancers, 2021, doi:10.3390/cancers13112624_

Round 1
Reviewer 1 Report
Dumitru et al. identified Tctex1 as a prognostic biomarker GBM and further described its biological functions through mediating cell proliferation, invasion, phosphorylation of RB and MMP2 release. This is an interesting study and prognostic effect of Tctex1 could have clinical significance. The manuscript has a clear structure and is well-written. Although these strengths, the manuscript suffers from several critical flaws.
- The first concern comes the quantification of Tctex1. To the reviewer’s point, the measurement of the expression level is subjective. Without objective measurement, the conclusion is questionable.
- The separation of patients with high vs low Tctex1 is arbitrary. Since Cox regression model is applied, the prognostic effect of Tctex1 can be evaluated in the form of continuous data (rather than the transformation of binary data), which is a more unbiased way.
- Multivariate analysis should be conducted by including all the confounding factors in one model. In this article, the authors performed 4 different models with each including only two predictors, which is inefficient and not a correct way to adjust all confounding variables.
- The fact of favorable prognostic effect of IDH1 mutation and MGMT methylation has been recognized. If the status of these two factors is available to the author, it should be included to strength the manuscript.
Minor comments:
1) Several typos are found. For example, line 114 “exhibited” should be “exhibited”?; line 115 the quote of weak, medium or strong should be corrected
Reviewer 2 Report
In principle, the role of Tctex1 is elucidated in this manuscript which is novel and really interesting. Nevertheless, detailed intracellular processes or interactions are not described. Graphical documentaion is clear and consice.
Major concerns:
- The authors compared 202 GBM samples with 42 samples of tumor free brain tissue. They did not state if the samples were paired for statistical testing. How did they pair 202 tumor tissues with 42 normal tissues? If samples are paired (which should be the approach) using U-test is not appropriate. Please provide more information and explain statitistical approach (Please refer to data of Figure 2 and 5).
- Does the H-score recognizes the cell density? In which cells Tctex1 is expressed in normal brain to be compared to malignant gial cells? Did you recognize the cell type specific expression? Please demonstrate expression in normal brain in an identifiable photograph (higher resolution).
- Please provide information of the MTT assay: number of replicates, statistical testing, usage of blank value (Figure 4 A,B).
- Please provide testing and p-value for clonogenic assay since you presented a significant difference (Figure 4 C, D). Are criteria for testing fulfilled (number of assays in each cell line)? Or does the last sentence refers to all experiments shown in Figure 4? If this is true how did you test normal distribution of 3 replicates? The same refers to data in Figure 6 C and F.
- GBM can be IDH mutant or wildtype, please identify this major feature in your cohort. Is expression of Tctex1 independent of IDH status? Please refer to Figure 2 C, D concerning survival and PFS (please integrate IDH status).
- Finally, please explain why it may be not adquate just to stain pRB in GBM samples because it correlates with Tctex1 expression nicely. RB is already known to be important in GBM. Please discuss what me be the molecular background of increased Tctex1 expression in GBM.
Round 2
Reviewer 1 Report
The manuscript has been improved a lot. I still have some comments:
- For figure 2E, the row of Tctex1low should be removed. HR of Tctex1 (high vs low) has been provided by Tctex1high and there is no point to show the reference Tctex1low variable. In addition, there are 7 independent variables in the Cox model. The proportional hazard assumption of Cox model should be evaluated. If it is already done, it should be included in the manuscript.
- Figure 5A, it should be more suitable to choose one of the comparisons: GBM(all) vs health brain, or GBM(matched) vs health brain. Showing all the three boxplots is confusion and it will be misunderstood as the comparison of 3 groups.
- Line 334, p=836?
Reviewer 2 Report
I have no further suggestions. The revision is adequate.
Author Response
Thank you for reviewing and improving our manuscript.